# AIMing for Standardised Explainability Evaluation in GNNs: A Framework and Case Study on Graph Kernel Networks

**Magdalena Proszewska**                                    *m.proszewska@ed.ac.uk*
*School of Informatics, University of Edinburgh*

**N. Siddharth**                                                *n.siddharth@ed.ac.uk*
*School of Informatics, University of Edinburgh*

**Reviewed on OpenReview:** *https://openreview.net/forum?id=onZkYXI7oe*

## Abstract

Graph Neural Networks (GNNs) have advanced significantly in handling graph-structured data, but a comprehensive framework for evaluating explainability remains lacking. Existing evaluation frameworks primarily involve post-hoc explanations, and operate in the setting where multiple methods generate a suite of explanations for a single model. This makes comparison of explanations across models difficult. Evaluation of inherently interpretable models often targets a specific aspect of interpretability relevant to the model, but remains underdeveloped in terms of generating insight across a suite of measures. We introduce **AIM**, a comprehensive framework that addresses these limitations by measuring **A**ccuracy, **I**nstance-level explanations, and **M**odel-level explanations. AIM is formulated with minimal constraints to enhance flexibility and facilitate broad applicability. Here, we use AIM in a pipeline, extracting explanations from inherently interpretable GNNs such as graph kernel networks (GKNs) and prototype networks (PNs), evaluating these explanations with AIM, identifying their limitations and obtaining insights to their characteristics. Taking GKNs as a case study, we show how the insights obtained from AIM can be used to develop an updated model, xGKN, that maintains high accuracy while demonstrating improved explainability. Our approach aims to advance the field of Explainable AI (XAI) for GNNs, providing more robust and practical solutions for understanding and improving complex models.

## 1 Introduction

Explainability in for Graph Neural Networks (GNNs) (Kipf & Welling, 2017; Kakkad et al., 2023) has primarily focused on post-hoc explanations—interpreting trained models by assigning importance scores to graph elements (Ying et al., 2019; Luo et al., 2020). These explainers are designed to probe models, post training, to highlight which parts of the graph-structured input influence predictions the most.

Explainability for inherently interpretable GNNs—those that are explicitly designed for interpretability—includes architectures that use information constraints such as attention mechanisms, e.g. Graph Attention Networks (Velickovic et al., 2018; Brody et al., 2022; Noravesh et al., 2025), as well as structurally constrained models like Graph Kernel Networks (GKNs) (Nikolentzos & Vazirgiannis, 2020; Cosmo et al., 2021) and Prototypical GNNs (Ragno et al., 2022; Zhang et al., 2022b). These methods aim to express decision-making processes through internal structures, e.g. attention scores, graph filters, or learned prototypes.

Existing metrics for XAI in GNNs are largely designed for post-hoc settings (Tan et al., 2022; Chen et al., 2022; Amara et al., 2022; Zheng et al., 2024; Longa et al., 2025). They typically assume that multiple explainers are applied to a single, fixed model, enabling evaluation through changes in the model's output at its logits. While appropriate for identifying explanations consistent across different explainers, this setup becomes inadequate for comparison of explanations across different models. Moreover, with inherently interpretable models such as GKNs or PNs, explainability is typically restricted to specific measures; e.g. just on which nodes are relevant for the graph filters or prototypes.

To address these shortcomings, we propose AIM, a general-purpose evaluation framework for GNN explainability that relaxes these assumptions. AIM draws on both graph-specific evaluation protocols and universal properties of good explanations that apply across domains and explanation types (Nauta et al., 2023), enabling fair and consistent comparison across different XAI methods for GNNs. In order to apply AIM across a broad range of models, we also develop mechanisms to extract explanations of different kinds (e.g. SHAPExplainer for instance-level explanations from GKNs) as necessary. We conduct a suite of experiments to demonstrate the utility of the AIM metrics, making a particular case study of GKNs, to highlight that insights drawn from the AIM measures can be used to improve the explainability of such models without sacrificing model performance (as demonstrated by our GKN model, XGKN). While the case study focusses on GKN framework, many of the underlying principles generalise naturally to other interpretable graph models, such as Prototypical GNNs.

Our contributions are summarised as follows:

1. AIM: an evaluation framework for GNN explainability, assessing **A**ccuracy, **I**nstance-level, and **M**odel-level explanations across different models with fewer constraints.
2. Developing SHAPEXPLAINER—a method to extract instance-level explanations from Graph Kernel Networks and Prototypical GNNs via SHAP propagation.
3. Assessment of existing inherently interpretable GNNs with respect to their XAI capabilities.
4. Developing XGKN, a Graph Kernel Network with improved explainability informed by AIM analyses.

## 2 Related Work

### 2.1 Metrics for GNN Explainability

Effective evaluation of a model's explainability—regardless of modality—requires considering properties such as correctness, consistency, and complexity of explanations. The Co-12 framework (Nauta et al., 2023) establishes 12 properties of explanation quality, providing a standardised foundation for assessing XAI.

For GNNs, when going beyond ground-truth-based evaluation, approaches include metrics such as fidelity, robustness, sufficiency and necessity—with most employing fidelity-related measures such as fidelity+ (also called sufficiency) and sparsity to assess explanations. Fidelity metrics quantify how prediction outcomes change when input graphs are perturbed according to the explanation, enabling quantitative assessment of post-hoc explainers applied to a fixed trained model. However, as highlighted by Zheng et al. (2024), these evaluations often neglect out-of-distribution (OOD) shifts which occur when input graph perturbations create subgraphs that fall outside the data distribution seen by the model during training. Table 1 provides an overview of existing GNN explainability evaluation frameworks and the measures they evaluate with. Sparsity as typically defined prioritises conciseness over faithfulness, potentially leading to explanations that omit critical features; we hence ignore sparsity in our framework.

The majority of approaches apply post-hoc methods to fixed models, limiting applicability to inherently interpretable models and cross-model comparisons. Evaluation based on a single fixed model permits meaningful logit-based scores for comparing explanations; however, these metrics become unreliable across different models due to varying decision boundaries and output scales.

We propose a comprehensive set of metrics—AIM—which assesses XAI capabilities across three different categories: Accuracy, Instance level, and Model level. These metrics build upon the characteristics outlined in the Co-12 framework (Nauta et al., 2023) and integrate widely used measures for post-hoc explainers, while relaxing some constraints to enable broader applicability, such as the requirement that multiple explainers must be applied to a single, fixed model Agarwal et al. (2022); Amara et al. (2022).

> Evaluations of GNN explainability typically focus on a limited set of properties and on comparing different explainers on the same models, limiting generalisability and potentially missing broader perspectives. AIM provides a unified framework that integrates instance- and model-level analyses to offer a more comprehensive assessment of explainability.

Table 1: Quantitative evaluation criteria used in recent GNN explanation papers. Symbols: ✓ = evaluated; ∼ = evaluated without accounting for out-of-distribution (OOD) issues; blank = not evaluated.

| | Instance-level | Model-level | Eval w/ GTs | Sufficiency | Necessity | Robustness | Consistency | Sparsity | Correctness | Redundancy | Cross-explainer | Cross-model |
|---|---|---|---|---|---|---|---|---|---|---|---|---|
| GNNExplainer (Ying et al., 2019) | ✓ | | ✓ | | | | | | | | ✓ | |
| PGExplainer (Luo et al., 2020) | ✓ | | ✓ | | | | | | | | ✓ | |
| SubgraphX (Yuan et al., 2021) | ✓ | | | ∼ | | | | ✓ | | | ✓ | |
| ReFine (Wang et al., 2021) | ✓ | | ✓ | ∼ | | | | | | | ✓ | |
| VGIB (Yu et al., 2021) | ✓ | | | ∼ | ∼ | | | ✓ | | | ✓ | ✓ |
| Zorro (Funke et al., 2022) | ✓ | | ✓ | ✓ | | | | ✓ | | | ✓ | ✓ |
| GStarX (Zhang et al., 2022a) | ✓ | | | ∼ | ∼ | | | ✓ | | | ✓ | |
| CF-GNNExplainer (Lucic et al., 2022) | ✓ | | ✓ | ∼ | | | | ✓ | | | ✓ | |
| CF$^2$ (Tan et al., 2022) | ✓ | | ✓ | ∼ | ∼ | | | | | | ✓ | |
| PAGE (Shin et al., 2022) | | ✓ | ✓ | | | | ✓ | ✓ | | | ✓ | |
| GFlowExplainer (Li et al., 2023) | ✓ | | ✓ | ∼ | | | | ✓ | | | ✓ | |
| RC-Explainer (Wang et al., 2023) | ✓ | | | ∼ | | ✓ | | | | | ✓ | |
| D4Explainer (Chen et al., 2023) | | ✓ | | ∼ | ∼ | | | ✓ | | | ✓ | |
| GraphXAI (Agarwal et al., 2022) | ✓ | | ✓ | ∼ | | ✓ | | ✓ | | | ✓ | |
| GraphFramEx (Amara et al., 2022) | ✓ | | ✓ | ∼ | ∼ | | | | | | ✓ | |
| GNNInterpreter (Wang & Shen, 2024) | | ✓ | | ∼ | | | | | | | ✓ | |
| Zheng et al. (2024) | ✓ | | | ✓ | ✓ | | | | | | | ✓ |
| Longa et al. (2025) | ✓ | | ✓ | ∼ | ∼ | | | | | | ✓ | ✓ |
| Lukyanov et al. (2025) | ✓ | | | ∼ | | ✓ | ✓ | ✓ | | | | ✓ |
| AIM (Ours) | ✓ | ✓ | ✓ | ✓ | ✓ | ✓ | ✓ | | ✓ | ✓ | ✓ | ✓ |

## 2.2 Extracting Explanations from GNNs

Evaluating XAI capabilities, beyond ground truth, considers explanations at two levels: instance-level, which explain individual predictions, and model-level, which reveal global concepts learned by the model.

Instance-level explainers typically generate importance maps over input graph elements—nodes, edges, or features—that identify the most influential components driving a single prediction (Ying et al., 2019; Luo et al., 2020). Model-agnostic methods often use masking or subgraph sampling to assess how perturbations affect outputs. Many also leverage SHAP values (Lundberg & Lee, 2017; Duval & Malliaros, 2021; Akkas & Azad, 2024) to quantify feature importance. Model-level explanations summarise concepts the model has learned over the entire dataset, often represented as graphs that capture patterns associated with specific classes or outputs (Yuan et al., 2020; Wang & Shen, 2024).

For GNNs with interpretable components such as attention mechanisms, implicit features like attention weights serve directly as relevance indicators (Velickovic et al., 2018). Other interpretable models like Graph Kernel Networks (GKNs) (Nikolentzos & Vazirgiannis, 2020; Cosmo et al., 2021; Feng et al., 2022) and prototype-based GNNs (Zhang et al., 2022b) typically lack dedicated explainers and have not been systematically evaluated for their XAI capabilities. Unlike standard message-passing GNNs, GKNs use non-standard operations over kernel responses, making them less amenable to standard explainers.

To address this, we propose a SHAP-based explainer tailored for GKNs that attributes importance to input elements using kernel similarity scores. While many model-agnostic explainers could, in principle, be applied to GKNs, our approach provides a simple and effective method that exploits their unique architecture. Although proposed for GKNs, the principles generalise and can also be adapted to prototype-based models.

> A variety of model-agnostic GNN explainers exist, but these aren't tailored to GKNs' unique structure.

## 2.3 Inherently interpretable GNNs

One class of interpretable GNNs incorporates information constraints through implicit mechanisms such as attention to highlight relevant nodes, or edges during message passing (Velickovic et al., 2018; Noravesh et al., 2025), or information bottlenecks to compress representations and filter irrelevant features (Wu et al., 2020; Yu et al., 2021). Such mechanisms inherently highlight the model's focus and how decisions are made.

Another class of interpretable GNNs incorporates structural constraints within their architecture. These models are the primary focus of our work because, despite their strong accuracy performance and valuable features—such as explicitly learned graph concepts—their explainability capabilities remain underdeveloped.

Graph Kernel Networks (GKNs) (Nikolentzos & Vazirgiannis, 2020; Cosmo et al., 2021; Feng et al., 2022) are inherently interpretable GNNs that learn trainable kernel filters against which input graphs are compared, producing similarity scores (kernel responses) that drive predictions. Unlike typical message-passing GNNs, GKNs avoid oversmoothing and oversquashing by relying on these kernel-based comparisons rather than iterative node aggregation. They are especially suited for tasks that rely graph structures like motifs and subgraphs, which play a key role in areas such as molecular graph analysis (Maziarz et al., 2024).

Prototypical GNNs share a closely related principle. They consist of a standard message-passing backbone combined with a prototype layer, where graph embeddings generated by the backbone are compared to learnable prototype embeddings. Similarity scores from this comparison inform the final prediction. While conceptually akin to GKNs, prototypical GNNs work with learned embeddings of graphs rather than explicit graph filters, and face interpretability challenges similar to those seen in prototype networks for images.

Both GKNs and prototypical GNNs achieve high predictive accuracy on popular classification tasks. However, despite their inherent interpretability, formal evaluation of their explainability capabilities remains limited. Current assessments mostly involve qualitative inspection of learned graph filters or visualisations of embeddings projected onto input space, without rigorous, systematic XAI evaluation.

> GKNs and prototype-based GNNs only include very limited evaluations of explainability.

## 3 Preliminaries

### 3.1 Graph Kernel Networks

Graph Kernel Networks (GKNs) can be defined as a composition of three functions: $f_{\text{sim}}$ to extract similarity scores between input subgraphs and trainable concepts (graph filters), $f_{\text{agg}}$ to aggregate them, and $f_{\text{pred}}$ to produce the final prediction.

Let $\mathcal{G} \in \mathbb{G}$ be an graph of size $n$. Let $\mathcal{G}_v$ be a $k$-hop neighbourhood of node $v \in \mathcal{G}$, $k \in \mathbb{N}_+$. Let $\mathcal{H}_1, ..., \mathcal{H}_m$ be the set of $m$ graph filters, $m \in \mathbb{N}_+$. Here, nodes in graphs are ordered. We denote by $v$, both a node $v \in \mathcal{G}$ and its index in $\mathcal{G}$ for notational simplicity. Function $f_{\text{sim}}$ represents the graph kernel module, with kernel $K : \mathbb{G} \times \mathbb{G} \to \mathbb{R}_+$, which extracts kernel responses (similarity scores) as

$$f_{\text{sim}}(\mathcal{G}) = [K(\mathcal{G}_v, \mathcal{H}_i)]_{\substack{v=1,...,n \\ i=1,...,m}} \in \mathbb{R}_+^{n \times m}. \tag{1}$$

Function $f_{\text{agg}} = (f_{\text{agg}}^{(1)}, \ldots, f_{\text{agg}}^{(m)})$ aggregates response for each graph filter $\mathcal{H}_1, ..., \mathcal{H}_m$, s.t. $f_{\text{agg}} \circ f_{\text{sim}}(\mathcal{G}) \in \mathbb{R}^m$. Function $f_{\text{pred}}$ is a neural network that predicts output, e.g. an MLP. Graph filters $\mathcal{H}_1, \ldots, \mathcal{H}_m$, along with the parameters of the node features encoder and the predictor $f_{pred}$, are trainable parameters of the network.

Existing GKNs exhibit slight variations in their approach. In Nikolentzos & Vazirgiannis (2020), RWGNN compares the entire input graph against graph filters. KerGNNs (Feng et al., 2022) are a more generalised framework that considers node-centered subgraphs, as described above. It further incorporates input node features, combined with kernel responses, which are passed to $f_{agg}$ to compute graph embeddings as averages or sums. Both models leverage the Random Walk Kernel (Vishwanathan et al., 2010) for its computational efficiency and differentiability, and define graph filters using continuous adjacency and feature matrices,

optimised via standard gradient descent. In contrast, Cosmo et al. (2021) allows the use of non-differentiable kernels and employs a Discrete Randomised Descent strategy for training. The graph filters have a discrete representation, and kernel responses are aggregated by summation. All three GKNs define $f_{\mathrm{pred}}$ as an MLP.

### 3.1.1 Random Walk Kernel

The Random Walk Kernel (Vishwanathan et al., 2010), widely used in GKNs by virtue of being a differentiable graph kernel, counts the number of walks that two graphs have in common. Let $\mathcal{G}, \mathcal{G}' \in \mathbb{G}$ be graphs. Since a random walk on the direct product graph $\mathcal{G}_\times = \mathcal{G} \times \mathcal{G}'$ is equivalent to performing simultaneous random walks on graphs $\mathcal{G}$ and $\mathcal{G}'$, the $P$-step random walk kernel can be defined as

$$K_{RW}(\mathcal{G}, \mathcal{G}') = \sum_{p=0}^{P} K_{RW}^p(\mathcal{G}, \mathcal{G}') = \sum_{p=0}^{P} S^T A_\times^p S, \tag{2}$$

where $A_\times$ is an adjacency matrix of $\mathcal{G}_\times$, and $S = XX'^T$, where $X$ and $X'$ represents node features of $\mathcal{G}$ and $\mathcal{G}'$, respectively. $S_{ij}$ corresponds to similarity between $i$-th node in $\mathcal{G}$ and $j$-th node in $\mathcal{G}'$.

### 3.2 Prototypical GNNs in relation to GKNs

Prototypical GNNs closely resemble GKNs in structure and can similarly be defined as $f_{pred} \circ f_{agg} \circ f_{sim}$. In ProtGNN (Zhang et al., 2022b), $f_{sim}$ samples input subgraphs, embeds them, and compares to learnable prototypes $\mathcal{H}_1, \ldots, \mathcal{H}_m$. Aggregation $f_{agg}$ uses max, and $f_{pred}$ is a linear layer followed by softmax. Prototypes are represented by embeddings that can be projected onto the input space to yield model-level explanations in a graph form. Other prototype-based GNNs (Dai & Wang, 2022) follow similar principles.

## 4 AIM metrics

We identify essential properties of GNN-based explanations and propose a principled set of metrics to evaluate them. We first clarify what we mean by an explanation to ensure comparability across methods and models. We differentiate between instance-level and model-level explanations. Instance-level explanations are represented as induced subgraphs from the input graph, highlighting the parts of the graph most relevant to a specific prediction. In contrast, model-level explanations are presented as graphs within the input space, capturing broader patterns learned by the model. These model-level explanations can be intrinsic to the model architecture, such as prototypes in Prototypical GNNs or graph filters in Graph Kernel Networks.

Explainability methods typically output continuous feature importance maps, which require thresholding to pinpoint the relevant parts. As noted earlier, varying models and graph structures lead to different decision boundaries and scales, making raw importance values challenging to compare directly. By applying common thresholding techniques and focusing on induced subgraphs, we standardise the representation of explanations across models, thereby enabling the definition of metrics that are applicable across approaches. While we assume the use of GNNs for graph classification tasks, our evaluation framework is general and can be adapted to other tasks with minor modifications to the metric definitions.

### 4.1 Properties to cover

Co-12 (Nauta et al., 2023) is a set of conceptual properties that are essential for a comprehensive assessment of explanation quality. Inspired by it, we propose AIM, a set of 10 metrics divided into 3 categories to assess: accuracy (A1-A2), instance-level explanations (I1-I5) and model-level explanations (M1-M3). These metrics capture 10 essential properties of GNN explanations, as outlined in Table 2.

AIM combines multiple complementary metrics to provide a comprehensive assessment of GNN explanations. When ground truth is available, we include evaluation against these references as a direct measure of explanation quality. However, such ground truth is often unavailable or incomplete in real-world settings, limiting their applicability. To address this, we emphasise metrics based on sufficiency and necessity, which assess whether the explanation contains enough information to preserve the model's prediction and whether

Table 2: Summary of Properties Covered by AIM, Based on the Co-12 Framework.

| Metric | Description and justification |
|---|---|
| A1: Accuracy (instance-level) | Instance-level explanations should match ground truths. Ensures that the model's predictions are accurately explained. |
| A2: Accuracy (model-level) | Model-level explanations should align with ground truths. Ensures that the model's global decision-making logic is accurately represented. |
| I1: Sufficiency | Explanations should include all relevant features necessary for the prediction. Ensures that explanations capture all critical factors that influence predictions. |
| I2: Necessity | Only the most critical features should be included in the explanation. Helps to focus on the most relevant information, improves clarity, reduces complexity. |
| I3: Robustness (nodes) | Explanations should remain stable under minor node perturbations. Ensures that the explanation is robust to irrelevant changes in node features, and therefore more reliable. |
| I4: Robustness (edges) | Explanations should remain consistent under minor edge perturbations. Ensures that the explanation is robust to irrelevant changes in the graph structure, and therefore more reliable. |
| I5: Consistency | Explanations should remain stable across different conditions. Ensures that minor variations, such as random initializations, do not cause significant changes in the explanations, leading to more reliable insights. |
| M1: Correctness (nodes) | Nodes variations in model-level explanations should reflect in instance-level explanations. Ensures alignment between global and local explanations, important for model transparency. |
| M2: Correctness (edges) | Edges variations in model-level explanations should reflect in instance-level explanations. Ensures alignment between global and local explanations, important for model transparency. |
| M3: Redundancy | Set of model-level explanations should avoid redundancy. Helps in creating concise, easier to interpret explanations, avoiding unnecessary complexity. |

all included features are essential, respectively. We further include robustness and consistency metrics to evaluate explanation stability under input perturbations and across different runs or models.

Crucially, our framework introduces a correctness metric to measure the alignment between model-level and instance-level explanations—leveraging the availability of both explanation types in our approach—and a redundancy metric to assess overlap within the set of model-level explanations. Although sparsity is commonly used to quantify explanation simplicity, we consider it a less reliable indicator of explanation quality. Sparsity favours conciseness but does not guarantee faithfulness or completeness, and may lead to misleadingly minimal explanations that omit important features. Therefore, while sparsity can aid interpretability, our evaluation prioritises metrics that better capture the explanatory power and reliability of the methods.

Beyond academic standardisation, AIM serves as a suite of sanity checks for real-world applications where XAI systems are deployed. By quantifying reliability through metrics that capture distinct and sometimes competing properties of explainability, the framework provides a rigorous diagnostic tool to identify systemic weaknesses. This enables a thorough pre-deployment evaluation, ensuring that a model's underlying logic is robustly and faithfully reflected in its practical decisions.

> AIM adapts established explainability properties (Co-12) for graphs, assessing explanations along multiple dimensions to reveal strengths and limitations.

## 4.2 AIM evaluation formulas

AIM framework builds on existing evaluation metrics for post-hoc XAI methods in GNNs (Bajaj et al., 2021; Tan et al., 2022; Chen et al., 2022; Agarwal et al., 2022; Amara et al., 2022; Zheng et al., 2024; Longa et al., 2025), with adjustments made to handle cross-model comparison. We formulate the AIM metrics as follows.

Let $\mathcal{G} \in \mathbf{D}$ be a graph from the dataset $\mathbf{D} \subset \mathbb{G}$. Let $X_{\mathcal{G}}$ and $A_{\mathcal{G}}$ denote its feature and adjacency matrices respectively. Let $f$ be the model we wish to explain. Let $h$ be an explainer, which for a graph $\mathcal{G}$ and prediction $f(\mathcal{G}) = c$ (class label), induces subgraph $h(\mathcal{G}) \subset \mathcal{G}$ as an explanation. Note that $h$ depends on $f$, and $h$ does not have to be deterministically defined, particularly when it requires training.

For subgraphs $\mathcal{G}_1, \mathcal{G}_2 \subseteq \mathcal{G}$, let $\mathrm{IoU}(\mathcal{G}_1, \mathcal{G}_2)$ denote the intersection over union of their nodes. Let $\mathbf{D}_X$ be the multiset of all node features present in the dataset $\mathbf{D}$. We define a feature perturbation function $\mathcal{P}_X(X_{\mathcal{G}}, \delta)$, which for each node $v \in \mathcal{G}$, takes its feature to be $X^v \sim \mathbf{D}_X$ with probability $\delta$, else keeps it as $X_{\mathcal{G}}^v$. Similarly, we define a perturbation function on adjacency matrices $\mathcal{P}_A(A_{\mathcal{G}}, \delta^+, \delta_-)$, which adds edges with probability $\delta^+$ and removes edges with probability $\delta^-$.

### 4.2.1 Instance-level

**A1: Accuracy (instance-level)** Compares the instance-level explanation $h(\mathcal{G})$ against the ground truth $\mathcal{E}$: $\mathrm{IoU}\big(h(\mathcal{G}), \mathcal{E}\big)$, where $\mathcal{E}$ represents the ground truth explanation for the considered task, $\mathcal{E} \subset \mathcal{G}$.

**I1: Sufficiency** Assesses consistency of the class prediction $c$ for subgraphs $\mathcal{S} \subset \mathcal{G}$ that contain the explanation $h(\mathcal{G})$: $\mathbb{1}[f(\mathcal{S}) = c]$, where $\mathcal{S} \sim \{\mathcal{S} \mid h(\mathcal{G}) \subset \mathcal{S} \subset \mathcal{G}\}$.

**I2: Necessity** Assesses change of the class prediction $c$ for subgraphs $\mathcal{S} \subset \mathcal{G}$ that do not contain nodes included in the explanation $h(\mathcal{G})$: $\mathbb{1}[f(\mathcal{S}) \neq c]$, where $\mathcal{S} \sim \{\mathcal{S} \mid \mathcal{S} \subset \mathcal{G}, h(\mathcal{G}) \not\subset \mathcal{S}\}$.

**I3: Robustness (nodes)** Evaluates change in the explanation $h(\mathcal{G})$ when the node features $X_{\mathcal{G}}$ of the input graph $\mathcal{G}$ are altered so that the class prediction $c$ does not change: $\mathrm{IoU}(h(\mathcal{S}), h(\mathcal{G}))$ for $\mathcal{S} \sim \mathcal{P}_X(\mathcal{G}, \delta)$ such that $f(\mathcal{S}) = c$, where $\delta \in (0, 1)$ is a hyperparameter.

**I4: Robustness (edges)** Evaluates the change in the explanation $h(\mathcal{G})$ when the edges of the input graph $\mathcal{G}$ are altered so that the class prediction $c$ does not change: $\mathrm{IoU}(h(\mathcal{S}), h(\mathcal{G}))$ for $\mathcal{S} \sim \mathcal{P}_A(\mathcal{G}, \delta^-, \delta^+)$ such that $f(\mathcal{S}) = c$, where $\delta^-, \delta^+ \in (0, 1)$ are hyperparameters.

**I5: Consistency** Measures consistency of the explainer $h$ on graph $\mathcal{G}$: $\mathrm{IoU}(h_1, h_2)$, where $h_1, h_2 \sim h(\mathcal{G})$.

Note that for the network $f$, explanation subgraphs determined by $h$ may fall outside the distribution, yielding various existing measures of sufficiency, necessity or fidelity (Pope et al., 2019; Yuan et al., 2022) ineffective. We aim to reduce the impact of the Out Of Distribution (OOD) issue by employing subgraph sampling technique based on random graph edits, following OOD analysis by Zheng et al. (2024).

### 4.2.2 Model-level

Let $f(\cdot|\theta, \{\mathcal{H}_i\}_{i=1}^m)$ be an interpretable GNN network to be investigated (Prototypical Network or Graph Kernel Network), where $\mathcal{H}_1, ..., \mathcal{H}_m$ denote graphs that represent identified concepts (projected prototypes or graph filters) and $\theta$ represents the rest of the model's parameters, $m \in \mathbb{N}_+$. Note that $\mathcal{H}_1, ..., \mathcal{H}_m$ are model-level explanations, and explainer function $h$ depends on $f(\cdot|\theta, \{\mathcal{H}_i\}_{i=1}^m)$.

**A2: Accuracy (model-level)** Compares model-level explanation $\mathcal{H}_1, \ldots \mathcal{H}_m$ against ground truths $\mathcal{E}_1, \ldots \mathcal{E}_k, k \in \mathbb{N}_+$: $\frac{1}{k}\sum_{j=1}^k \min_{i=1,\ldots m} \mathrm{GED}(\mathcal{H}_i, \mathcal{E}_j)$, where $\mathcal{E}_1, ..., \mathcal{E}_k$ are graphs that are ground truth model-level explanations, $k \in \mathbb{N}_+$, and GED denotes normalised graph edit distance.

**M1: Correctness (nodes)** Evaluates changes in the instance-level explanations $h(\mathcal{G})$, when node features in model-level explanations $\mathcal{H}_1, \ldots \mathcal{H}_m$ are altered: $\frac{1}{|\mathbf{D}|}\sum_{\mathcal{G} \in \mathbf{D}} \mathrm{IoU}(h(\mathcal{G}), h'(\mathcal{G}))$, where explainer function $h'$ explains $f(\cdot|\theta, \{\mathcal{H}_i'\}_{i=1}^m)$ such that $\mathcal{H}_i' \sim \mathcal{P}_X(\mathcal{H}_i, \delta)$, where $\delta \in (0, 1)$ is a hyperparameter.

**M2: Correctness (edges)** Evaluates changes in the instance-level explanations $h(\mathcal{G})$, when edges in model-level explanations $\mathcal{H}_1, \ldots \mathcal{H}_m$ are altered: $\frac{1}{|\mathbf{D}|}\sum_{\mathcal{G} \in \mathbf{D}} \mathrm{IoU}(h(\mathcal{G}), h'(\mathcal{G}))$, where explainer function $h'$ explains $f(\cdot|\theta, \{\mathcal{H}_i'\}_{i=1}^m)$ such that $\mathcal{H}_i' \sim \mathcal{P}_A(\mathcal{H}_i, \delta^-, \delta^+)$, where $\delta^-, \delta^+ \in (0, 1)$ are hyperparameters.

**M3: Redundancy** Assesses correlation between similarity scores of all graphs in the dataset $\mathbf{D}$ with respect to different concepts $\mathcal{H}_1, \ldots, \mathcal{H}_m$: $1/(m(m-1))\sum_{i=1}^m \sum_{j=i+1}^m \mathrm{CORR}(\{\varphi_i(\mathcal{G})\}_{\mathcal{G} \in \mathbf{D}}, \{\varphi_j(\mathcal{G})\}_{\mathcal{G} \in \mathbf{D}})$, where $\varphi_i(\mathcal{G}) = f_{agg}^{(i)} \circ f_{sim}(\mathcal{G}|\theta, \{\mathcal{H}_i\}_{i=1}^m) \in \mathbb{R}^m$, $i = 1, \ldots, m$, and CORR denotes the absolute Spearman rank correlation.

Each metric ranges from 0 to 1. A1 and I1–I5 are maximised; A2 and M1–M3 are minimised but reported as $1 - \gamma$ (with $\gamma$ being the original value) so higher scores always indicate better performance.

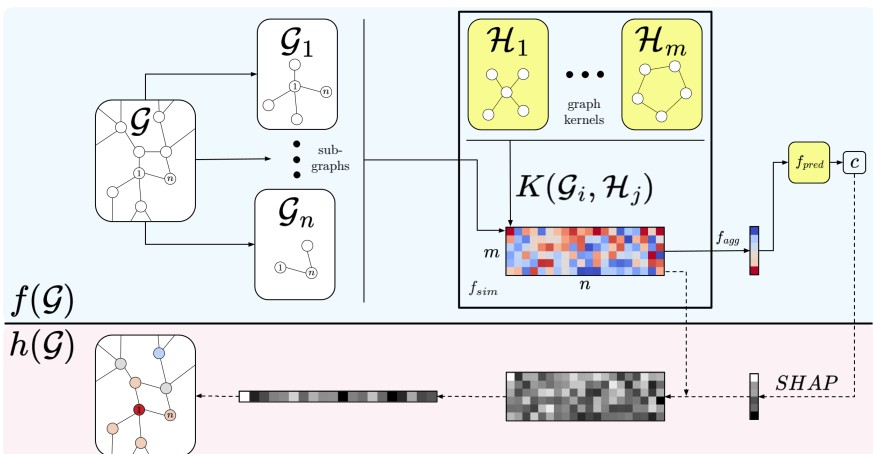

Figure 1: Overview of a GKN with SHAPExplainer. Top : forward pass. Bottom : explanation extraction. Learnable components in yellow. Input $\mathcal{G}$ (size $n$) is processed as a set of node-centered subgraphs $\mathcal{G}_1, ..., \mathcal{G}_n$. Similarity scores $f_{sim}$ are computed between subgraphs and kernels $\mathcal{H}_1, ..., \mathcal{H}_m$ using function $K$. Scores aggregated using $f_{\text{agg}}$ and class $c$ is predicted using predictor $f_{\text{pred}}$. For explanation extraction, SHAP values for $f_{\text{pred}}$ are propagated back onto the input graph $\mathcal{G}$ by reversal of score aggregation.

> AIM reduces assumptions to support fair comparison across models and explanation methods.

## 5   SHAPExplainer for GKNs

Inherently interpretable GNNs, such as Graph Kernel Networks (GKNs) or Prototypical Networks, identify graph concepts that drive predictions, which can be considered model-level explanations. Extracting instance-level explanations, however, is a more complex task.

Post-hoc explainers extract explanations from trained GNNs. Explainers can be applied to inherently interpretable GNNs, as long as they meet certain conditions, such as employing Message Passing (Luo et al., 2020)—a requirement not met by GKNs—or being fully differentiable (Ying et al., 2019), which GKNNs (Cosmo et al., 2021) are not. Some explainers do not require these conditions (Kokhlikyan et al., 2020).

We argue that designing an explainer for GKNs, with additional applicability to Prototypical GNNs, is a more effective approach. In particular, existing explainers operate on nodes and edges, which for GKNs must be converted to subgraphs, introducing a time-consuming overhead. We leverage the inherent structure of GKNs and propose a SHAP-based approach (Lundberg & Lee, 2017) to extract instance-level explanations. This approach is similar to GAT-specific explainers that interpret decisions by analysing attention scores.

As in Section 3, let $f = f_{pred} \circ f_{agg} \circ f_{sim}$ represent an interpretable GNN, where $f_{sim}$ yields similarity scores between input subgraphs and concepts, $f_{agg}$ aggregates scores over subgraphs, and $f_{pred}$ does the final prediction. Again, we focus on graph classification, but the approach extends to other use cases.

Let $\mathcal{G}$ be an input graph, let $m \in \mathbb{N}_+$ be the number of concepts identified by the network $f$. Let $S = f_{sim}(\mathcal{G}) \in \mathbb{R}^{n \times m}$, where $n$ denotes number of input subgraphs that are compared against concepts, $z = f_{agg}(S) \in \mathbb{R}^m$, and $p = f_{pred}(z) \in \mathbb{R}^c$, where $c$ is the number of classes, $c \in \mathbb{N}_+$. Let $\hat{p}$ be the logit predicted for the class that $\mathcal{G}$ will be classified as: $\hat{p} = \max_{i=1,...,c} p_i$.

First, we compute the SHAP values for the function $f_{pred}$, with input $z$ and prediction $\hat{p}$, yielding a set of values $\{\phi_i\}_{i=0}^m$, where $\sum_{i=0}^m \phi_i = \hat{p}$. Here, $\phi_0$ represents the expected value, while $\phi_i$ for $i = 1, \ldots m$ corresponds to the importance of each respective concept. For simplicity, we assume that the aggregation function $f_{agg}$ is a summation over subgraphs, $z_i = \sum_{j=1}^n S_{ji}$, although the framework can be extended to other aggregation schemes (see Appendix A.1). We define a map of importance over input subgraphs $w \in \mathbb{R}^n$ such that $w_j = \sum_{i=1}^m (\phi_i \cdot S_{ji})/z_i$, and hence $\phi_0 + \sum_{j=1}^n w_j = \hat{p}$.

For GKNs, input subgraphs are defined as the neighbourhoods of individual nodes. Assuming that nodes are ordered, let $v \in \mathcal{G}$ be the $i$-th node in graph $\mathcal{G}$, $i = 1, \ldots, n$. Then $\mathcal{G}_i$ represents the subgraph centered around $v$. The importance of node $v$ is then defined as $\psi(v) = w_i$. For Prototypical Networks, input subgraphs are sampled to maximise each similarity score. Let $\mathcal{G}_i$ represent input subgraph that is associated with similarity score $z_i$, $i = 1, \ldots, m$. We define importance of a node $v \in \mathcal{G}$ as $\psi(v) = \sum_i w_i$ for $i : v \in \mathcal{G}_i, i = 1, ..., m$.

The final map of nodes importance is defined as softmax$(\{\psi(v)\}_{v \in \mathcal{G}})$. To identify the set of important nodes and, subsequently, the induced subgraph of $\mathcal{G}$ that serves as the explanation, thresholding techniques have to be applied. Figure 1 shows an overview of a GKN with SHAPExplainer.

In our experiments, we show that the SHAPExplainer not only generates more accurate instance-level explanations for Graph Kernel Networks (GKNs) than popular baselines, but also outperforms them in terms of speed by avoiding the preprocessing overhead of converting graphs to subgraphs.

> SHAPExplainer leverages the structure of Graph Kernel Networks and propagates kernel responses onto the input graph to provide explanations.

## 6 Experiments

**Models** We evaluate XAI capabilities of Graph Kernel Networks (GKNs)—specifically, KerGNN (Feng et al., 2022) and GKNN (Cosmo et al., 2021)—and compare them against a set of baselines. The Prototypical Network, ProtGNN (Zhang et al., 2022b), is selected for its structural and interpretability similarities to GKNs. We also include GIN (Xu et al., 2019), a widely-used standard GNN model, as a reference point for general GNN behaviour. Additionally, we evaluate GAT (Velickovic et al., 2018), which provides inherent interpretability and is part of the broader family of models leveraging attention mechanisms. These baselines represent a range of models, each with different structures and varying levels of interpretability, showing that AIM can be applied across different approaches. Appendix A.3 compares key features of our baselines.

**Explainers** To extract instance-level explanations, we consider a set of post-hoc explainer methods: GNNExplainer (Ying et al., 2019), ShapleyValueSampling (Štrumbelj & Kononenko, 2010; Kokhlikyan et al., 2020), IntegratedGradients (Sundararajan et al., 2017; Kokhlikyan et al., 2020), an AttentionExplainer, and proposed SHAPExplainer, to identify the most suitable one for each model.

**Datasets** We use 6 popular datasets: (1) synthetic graphs: BA2Motifs (Luo et al., 2020) and BAMultiShapes (Azzolin et al., 2023), (2) molecular graphs: MUTAG (Debnath et al., 1991) and PROTEINS (Borgwardt et al., 2005), and (3) social graphs: IMDB-BINARY and IMDB-MULTI (Yanardag & Vishwanathan, 2015). Following common practice, we omit node features in synthetic datasets; GAT is an exception, requiring them for good accuracy. In social graphs, node degree is used as the feature if model allows for it (all except GKNN). BA2Motifs, BAMultiShapes and MUTAG provide ground truth explanations

**Metrics** We evaluate prediction accuracy and proposed AIM metrics. With multiple metrics, identifying the best model is a complex task. AIM is designed to highlight particular areas where the model faces challenges in terms of XAI, rather than just providing a single, overarching performance score. We use a two-sided t-test with a significance level 0.05 to identify and highlight the top-performing models and explainers.

**Hyperparameters** We select hyperparameters based on the authors' guidelines and optimise them for the best predictive accuracy. For calculating SHAP, we use Deep SHAP (Lundberg & Lee, 2017). Hyperparameters chosen for the AIM metrics are provided in the Appendix A.2.

**Thresholding** We normalise importance maps to sum to one and select a percentile threshold $p \in [0, 1]$ so that cumulative importance below it equals $p$. Nodes scoring above this threshold are deemed relevant. For each experiment, $p \in \{0.1, 0.2, \ldots, 0.9\}$ is chosen to maximise A1 (if ground truth is available) or I1+I2 (see Appendix A.4 for sensitivity analysis regarding the choice of $p$).

Code to reproduce experiments is available at https://github.com/mproszewska/aim-xgkn.

Table 3: Evaluation across models and explainers. Bolding indicates statistically comparable or better scores.

(a) Instance-level accuracy (A1) and running times.

| Model | Explainer | BA-2motifs | | MUTAG | |
|---|---|---|---|---|---|
| | | A1 | Time (s) | A1 | Time (s) |
| GIN | GNNExplainer | 0.25±0.06 | 0.27±0.00 | **0.77±0.02** | 0.02±0.00 |
| | IntegratedGradients | **0.50±0.10** | 0.06±0.00 | **0.77±0.03** | 0.01±0.00 |
| | ShapleyValueSampling | 0.41±0.13 | 1.80±0.02 | 0.70±0.07 | 0.89±0.03 |
| GAT | GNNExplainer | 0.20±0.01 | 0.03±0.00 | **0.73±0.11** | 0.04±0.00 |
| | IntegratedGradients | **0.26±0.09** | 0.01±0.00 | **0.73±0.10** | 0.01±0.00 |
| | AttentionExplainer | 0.20±0.01 | 0.00±0.00 | **0.75±0.02** | 0.01±0.00 |
| | ShapleyValueSampling | **0.24±0.05** | 0.24±0.03 | 0.70±0.20 | 1.53±0.05 |
| ProtGNN | GNNExplainer | 0.18±0.02 | 0.06±0.00 | **0.76±0.02** | 0.07±0.00 |
| | IntegratedGradients | **0.31±0.12** | 0.01±0.00 | 0.64±0.02 | 0.01±0.00 |
| | ShapleyValueSampling | **0.32±0.05** | 3.60±0.01 | 0.65±0.00 | 1.76±0.08 |
| | SHAPExplainer | **0.33±0.03** | 47.44±1.42 | 0.64±0.02 | 21.56±3.21 |
| GKNN | ShapleyValueSampling | 0.20±0.01 | 23.26±0.08 | 0.54±0.06 | 8.15±0.44 |
| | SHAPExplainer | **0.49±0.04** | 0.19±0.00 | **0.79±0.03** | 0.06±0.03 |
| KerGNN | GNNExplainer | 0.26±0.04 | 2.32±0.01 | **0.78±0.02** | 1.30±0.04 |
| | IntegratedGradients | **0.33±0.07** | 0.59±0.01 | 0.72±0.06 | 0.35±0.01 |
| | ShapleyValueSampling | **0.32±0.07** | 26.13±0.06 | 0.75±0.03 | 9.84±0.59 |
| | SHAPExplainer | **0.33±0.12** | 0.06±0.00 | **0.76±0.03** | 0.01±0.00 |

(b) Predictive accuracy scores.

| | BA-2motifs | BAMultiShapes | MUTAG | PROTEINS | IMDB-B | IMDB-M |
|---|---|---|---|---|---|---|
| GAT | 92.4±6.4 | 90.0±9.7 | 80.0±9.9 | **73.7±3.5** | 69.8±3.7 | **45.1±3.2** |
| GIN | **99.6±0.7** | **94.5±3.0** | 81.5±9.8 | **75.4±4.1** | 70.0±4.2 | **45.0±5.1** |
| ProtGNN | **100.0±0.0** | 85.0±7.5 | **89.4±7.2** | 66.1±7.4 | **78.3±5.8** | 45.2±8.5 |
| GKNN | **99.2±0.2** | 90.7±7.9 | 75.7±6.9 | 68.5±6.6 | 61.1±9.4 | 44.1±4.2 |
| KerGNN | **99.4±0.1** | **90.4±4.9** | 81.9±6.0 | 70.7±8.0 | 71.1±7.4 | 43.7±4.5 |

(c) Instance-level (A1) and model-level (A2) accuracies.

| | BA-2motifs | | BAMultiShapes | | MUTAG | |
|---|---|---|---|---|---|---|
| | A1 | A2 | A1 | A2 | A1 | A2 |
| GIN | **0.51±0.11** | – | **0.21±0.06** | – | **0.78±0.03** | – |
| GAT | 0.26±0.10 | – | 0.17±0.05 | – | 0.73±0.06 | – |
| ProtGNN | 0.33±0.06 | 0.76±0.02 | **0.22±0.07** | 0.80±0.02 | 0.65±0.02 | 0.75±0.03 |
| GKNN | 0.49±0.04 | **0.80±0.01** | 0.17±0.11 | **0.83±0.01** | **0.80±0.03** | 0.77±0.01 |
| KerGNN | 0.33±0.12 | 0.68±0.01 | 0.10±0.06 | 0.72±0.01 | 0.76±0.04 | **0.80±0.01** |

## 6.1 Pairing models with explainers

First, we identify the best explainer for each model using BA2Motifs and MUTAG, which provide ground truth explanations. Selection is based on how well instance-level explanations align with these ground truths (A1). Additionally, we report time needed to extract explanations, highlighting how certain approaches are not most efficient for GKNs that require graph to subgraphs conversion. In cases when the results are relatively similar, we choose to use a more time-efficient explainer. Results are presented in Table 3a.

Note that the long running time for ProtGNN with SHAPExplainer is due to its input subgraph sampling mechanism. Therefore, we opt to use different explainer (ShapleyValueSampling) for that model. For GIN and GAT, we use IntegratedGradients. For the remaining GKNs, we choose SHAPExplainer.

> SHAPExplainer produces better and faster explanations from GKNs than other compatible explainers.

## 6.2 AIM evaluation

**Predictive accuracy** Table 3b shows that all models achieve comparable predictive accuracy.

**Explanations accuracy (A1-A2)** We first evaluate A1–A2 metrics using datasets that offer ground-truth explanations. Table 3c shows that GKNN achieves explanation accuracy comparable to GIN, despite GIN being a standard black-box model paired with a strong post-hoc explainer. In contrast, other inherently interpretable models perform worse in this setting. Results suggest that simply incorporating interpretable components does not guarantee good explanations. GKNN offers competitive performance with built-in model-level explanations but suffers from limited scalability due to non-differentiable components. KerGNN, while less accurate, shares a similar structure and, as we argue, can be refined for stronger XAI performance.

Evaluation of the remaining AIM metrics is shown in Figure 2.

**Instance-level explanations (I1-I5)** GIN and GAT achieve similar results, but GIN offers better explanations despite GAT's attention mechanism intended to improve interpretability. ProtGNN's explanations are less robust (I3–I4) than GKNs', likely due to variability from subgraph sampling. However, it outperforms on sufficiency and necessity (I1–I2), indicating better capture of essential features. Notably, all models—GKNs and ProtGNN—exhibit low consistency (I5), especially on more complex, non-synthetic tasks, which reflects the inherent challenge of producing stable explanations under subtle internal or environmental variations.

**Model-level explanations (M1-M3)** We observe that model-level explanations often fail to reflect changes in instance-level explanations—modifying model-level concepts does not lead to substantial changes at the instance level (M1-M2). This suggests that the models might not be utilising these learned concepts as effectively as intended. Moreover, we see a significant redundancy in the learned explanations (M3), indicating that many concepts may overlap or convey similar information.

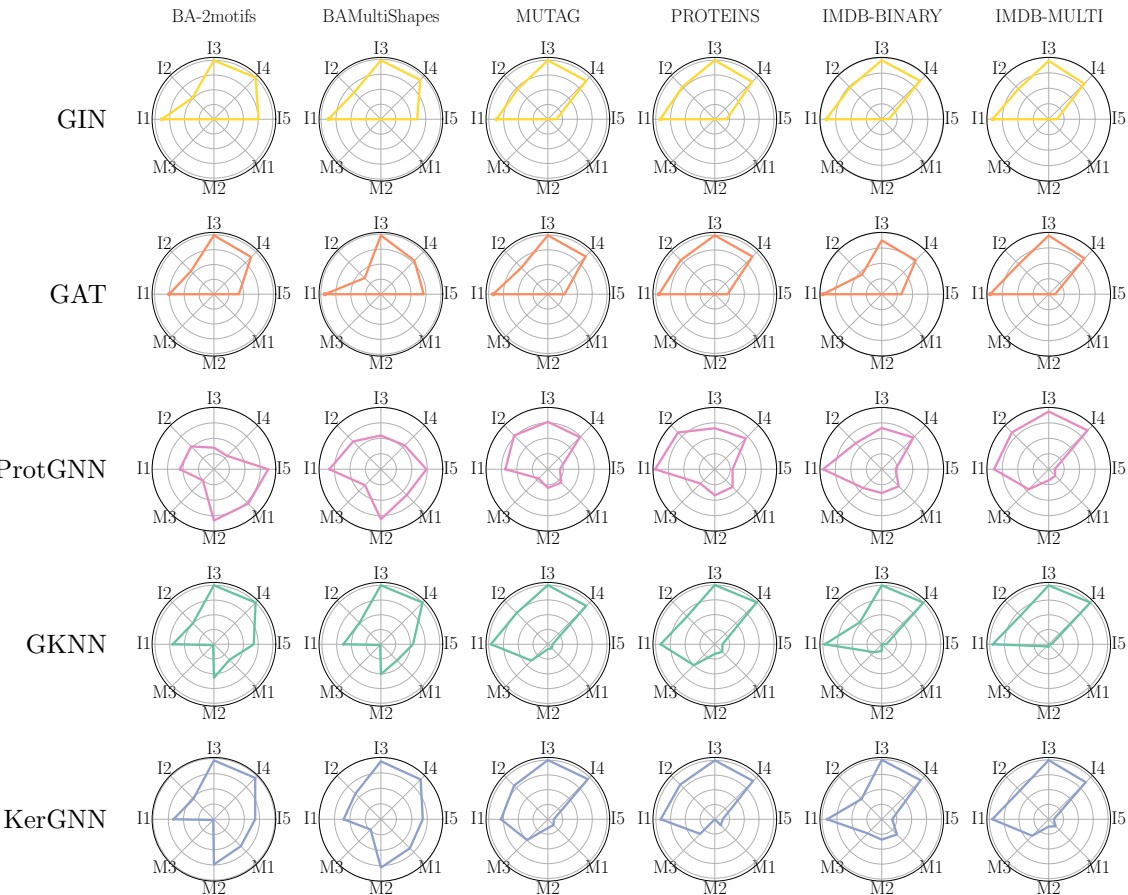

Figure 2: AIM metrics measured for GIN, GAT, Prototypical Network (ProtGNN), and Graph Kernel Networks (KerGNN, GKNN). Metrics have been oriented such that higher values indicate better performance.

While ProtGNN offers strong XAI capabilities, it has limitations, particularly in terms of robustness. We recognise the conceptual advantages of GKNs, in particular KerGNN, especially their full differentiability, which influences scalability. In the following section, we introduce a model that builds on the principles of KerGNN and other GKNs, aiming to improve explainability by: (1) extracting more relevant concepts, and (2) simplifying the differentiation between relevant and irrelevant nodes in importance maps.

> KerGNN has limited XAI; we argue it can improve, supported by less scalable GKNN's performance.

## 6.3 XGKN

Following prior notation, we define the network XGKN as a composition of three functions $f_{pred} \circ f_{agg} \circ f_{sim}$, where $f_{sim}$ extracts kernel responses, $f_{agg}$ aggregates them, and $f_{pred}$ produces final prediction.

As in GKNN and KerGNN, $f_{sim}$ compares $k$-hop neighbourhoods $\mathcal{G}_v$, $v \in \mathcal{G}$ against graph filters. We want to associate kernel responses for $\mathcal{G}_v$ with the importance of node $v$, hence we define the kernel $K$ as a Random Walk Kernel in which only walks that start in $v$ are counted. To improve the fairness of the comparison of responses from different filters, we normalise feature embeddings. For filter $\mathcal{H}_i$, $i = 1, \dots m$, we obtain

$$K(\mathcal{G}_v, \mathcal{H}_i) = \sum_{p=0}^{|\mathcal{H}_i|} \sum_{\substack{u \in \mathcal{G}_v \\ v', u' \in \mathcal{H}_i}} (S^T A_\times^p S)_{(v,v'),(u,u')}, \tag{3}$$

Table 4: XGKN results. Bold values indicate scores that are comparable or better than analogous results.

(a) A1 scores and running times (analogous to Table 3a).

| Explainer | BA-2motifs | | MUTAG | |
|---|---|---|---|---|
| | A1 | Time (s) | A1 | Time (s) |
| GNNExplainer | 0.16±0.01 | 6.55±0.02 | 0.77±0.02 | 6.44±0.54 |
| IntegratedGradients | 0.10±0.03 | 1.59±0.00 | 0.70±0.09 | 1.53±0.06 |
| ShapleyValueSampling | 0.11±0.02 | 71.87±1.80 | 0.73±0.11 | 38.61±2.03 |
| SHAPExplainer | **0.50±0.10** | 0.04±0.01 | **0.80±0.01** | 0.01±0.00 |

(b) Predictive accuracy (analogous to Table 3b).

| BA-2motifs | BAMultiShapes | MUTAG | PROTEINS | IMDB-B | IMDB-M |
|---|---|---|---|---|---|
| **100±0** | **92.8±4.0** | 84.2±6.9 | **74.5±2.1** | 71.3±7.6 | **45.6±5.4** |

(c) A1 and A2 scores (analogous to Table 3c).

| BA-2motifs | | BAMultiShapes | | MUTAG | |
|---|---|---|---|---|---|
| A1 | A2 | A1 | A2 | A1 | A2 |
| **0.50±0.11** | **0.77±0.13** | **0.31±0.09** | 0.78±0.00 | **0.80±0.02** | **0.81±0.01** |

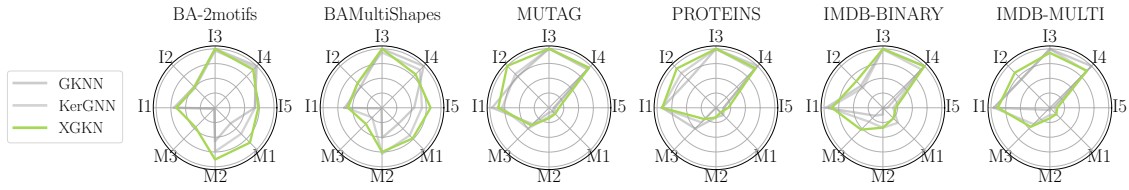

Figure 3: AIM evaluation of XGKN.

where $A_\times$ is an adjacency matrix of $\mathcal{G}_v \times \mathcal{H}_i$, and $S = X_{\mathcal{G}_v} X_{\mathcal{H}_i}^T$, where $X_{\mathcal{G}_v}$ and $X_{\mathcal{H}_i}$ represent normalised (encoded) node features of $\mathcal{G}_v$ and $\mathcal{H}_i$, respectively. We define

$$f_{sim}(\mathcal{G}) = [K(\mathcal{G}_v, \mathcal{H}_i)]_{\substack{v=1,\ldots,n \\ i=1,\ldots,m}} \in \mathbb{R}_+^{n \times m}. \tag{4}$$

Let $R = f_{sim}(\mathcal{G})$. Instead of using the default summation as the aggregation function $f_{agg}$, we opt to normalise $R$ and take the negative entropy to capture the relative contributions of each node and graph filter. We define the aggregation function $f_{agg} = \left( f_{agg}^{(1)}, \ldots, f_{agg}^{(m)} \right)$,

$$f_{agg}^{(i)}(R) = \sum_{v=1}^n \frac{R_{vi}}{\| R \|} \log \frac{R_{vi}}{\| R \|} \in \mathbb{R}, \quad i = 1, \ldots, m. \tag{5}$$

For the predictor $f_{pred}$, we employ a single linear layer or MLP, and aim to use as little layers as possible to achieve desired accuracy. This setup facilitates optimisation, encourages the model to learn a more effective set of graph filters, and helps prevent overly complex dependencies between them and final predictions. Our network, like KerGNN, is fully differentiable. The graph filters $\mathcal{H}_1, \ldots, \mathcal{H}_m$, along with the parameters of the predictor function $f_{pred}$, are parameters of the network optimised during training using gradient descent.

The use of feature embedding normalisation, combined with the negative entropy term, encourages the model to learn a more diverse and expressive set of graph filters. This design choice is expected to enhance model-level explanation quality, as reflected in metrics M1, M2, and M3. In addition, the kernel function based on random walk paths from a given node is intended to help the model more effectively distinguish relevant from irrelevant nodes, thereby improving instance-level interpretability—particularly metric I2 (necessity).

**Hyperparameters** For GKN-specific hyperparameters, we perform a grid search considering those best suited for GKNs (Section 6). These include: number of graph filters (4, 8, 16), size of graph filters (6, 8), node embeddings size (16), radius of node-centered subgraphs (2, 4) and their max size (10, 20). We use a single-layer classifier, preceded by batch normalisation, to make the final prediction. We train for up to 1000 epochs using the Adam optimiser, a learning rate of 0.01, weight decay of 1e-4, and batch size of 64.

**Results** Table 4a shows our SHAPExplainer achieves the most accurate instance-level explanations, demonstrating synergy with our model. Table 4b reports predictive accuracies. Table 4c shows notable improvements in explanation accuracy metrics, confirming the effectiveness of our approach. Finally, Figure 3 illustrates enhancements across a broader range of AIM metrics, showcasing our model's advancement in explainability. Tables with all results combined are in the Appendix A.5.

> XGKN explanations are more accurate, better at isolating necessary information, and show improved alignment between model- and instance-level explanations compared to previous GKNs.

## 7 Conclusions

In this work, we introduced AIM, a comprehensive evaluation framework for explainability in Graph Neural Networks (GNNs) that unifies accuracy, instance-level and model-level metrics. AIM enables consistent, cross-model assessment of explanation quality, filling a gap in current XAI evaluation practices.

We introduced a SHAP-based explanation method designed for Graph Kernel Networks (GKNs)—a class of inherently interpretable models that learn structured, graph-based concepts. Using the AIM framework, we evaluated the explainability of GKNs and identified their limitations. To address these shortcomings, we proposed XGKN, an enhanced variant of GKN, and demonstrated that it delivers improved explainability.

This work introduces a systematic framework for comparing the XAI capabilities of diverse GNN methods and makes a case for advancing architectures like GKNs, which embed interpretability into their design.

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

# A Appendix

## A.1 SHAP Explainer under Different Aggregation Functions

We now describe how attributions are propagated under different aggregation mechanisms. When max pooling is used (e.g., in ProtGNN (Zhang et al., 2022b)), the aggregated representation satisfies $z_i = S_{ji}$ for the maximising index $j$. Since only the element attaining the maximum contributes to the output in dimension $i$, the attribution $\phi_i$ is assigned entirely to that element. Formally, we set $w_j = \phi_i$, while all other elements receive zero contribution for this dimension.

For negative entropy aggregation, using the notation introduced in Section 5, the score can be expressed as

$$S_{ji} = \frac{R_{ji}}{\|R\|} \log \frac{R_{ji}}{\|R\|},$$

which decomposes additively across components. This additive structure allows us to propagate attributions in the same manner as in the sum-aggregation case, distributing contributions proportionally according to the component-wise decomposition.

## A.2 Hyperparameters for AIM evaluation

For AIM evaluation, certain perturbations are performed on input graphs (I1-I4), model's concept parameters are altered (M1-M2). Here, we specify hyperparameters for these operations.

For I1, we sample subset of nodes to be included in the induced subgraph along with nodes that are part of explanation. Each node is included with probability 0.5. For I2, we sample subset of nodes to be included in the induced subgraph excluding nodes that are part of the explanation. Each node is included with probability 0.5. For I3, we sample subset of nodes, except of those that are part of the explanation, which features are altered. Each node is altered with probability 0.1. New features are sampled from the dataset. For I4, we sample subset of edges, except of those that are part of the explanation, which are altered. Each edge is removed with probability 0.1, new edges are created with probability 0.1 multiplied by the average density a graph in the dataset. For M1, concept modification involves changing the features of each node with probability 0.5, assigning random features from the dataset, and encoding them if required. For M2, concepts are perturbed by either adding a non-existent edge or removing an existing edge between node pairs, each with probability 0.5.

## A.3 Baselines

Table 5 summarises the baseline models used in our study.

Table 5: Similarities and differences among the baseline models, highlighting the advantages of KerGNNs, which combine differentiability with interpretable, explicitly structured graph-based representations.

| Model | Message Passing | Interpretable component | Representation of interpretable components | How to interpret | Graphs compared against concepts | Comparison function |
|-------|-----------------|------------------------|-------------------------------------------|------------------|---------------------------------|---------------------|
| GIN | Yes | None | N/A | N/A | N/A | N/A |
| GAT | Yes | Attention Mechanism | Attention Scores | Scores indicate inter-node influence | N/A | N/A |
| ProtGNN | Yes | Prototypes | Embeddings | Project embeddings onto input space | Full graph or sampled subgraphs | Embeddings similarity |
| GKNN | No | Graph Filters | Discrete node features + adjacency matrix | Filters are explicit graph structures | Node-centered subgraphs | Various graph kernels (non-differentiable) |
| KerGNN | No | Graph Filters | Continuous node features + adjacency matrix | Filters are explicit graph structures | Node-centered subgraphs | Random Walk kernel (differentiable) |

### A.4 Sensitivity analysis of thresholding

To investigate the impact of the thresholding parameter $p$ used to induce explanatory subgraphs, we perform a sensitivity analysis. We evaluate the performance of both the KerGNN and XGKN models by perturbing the optimal threshold $p$ by $\pm 0.1$, and observe changes in AIM scores.

As shown in Table 6 and Figure 4, the results demonstrate that the explainability scores are quite consistent. Overall, shifting the threshold leads to slightly lower or nearly identical performance, confirming that the initial selection was robust. An unpaired t-test revealed no statistically significant differences in scores across tested thresholds ($p$, $p - 0.1$, and $p + 0.1$). This stability suggests that the induced subgraphs are not overly sensitive to minor hyperparameter adjustments.

Table 6: Instance-level (A1) accuracies for varying threshold, averaged over 10 splits with standard deviation. The value $p$ denotes the threshold selected by our default procedure (see Section 6). Note that model-level accuracy (A2) is independent of the threshold.

|  |  | BA-2motifs | BAMultiShapes | MUTAG |
|---|---|---|---|---|
| KerGNN | $p$ | 0.331±0.121 | 0.095±0.055 | 0.763±0.035 |
|  | $p - 0.1$ | 0.264±0.030 | 0.093±0.029 | 0.761±0.053 |
|  | $p + 0.1$ | 0.256±0.030 | 0.092±0.044 | 0.689±0.106 |
| XGKN | $p$ | 0.503±0.110 | 0.309±0.089 | 0.800±0.019 |
|  | $p - 0.1$ | 0.490±0.073 | 0.218±0.046 | 0.790±0.028 |
|  | $p + 0.1$ | 0.379±0.058 | 0.199±0.067 | 0.770±0.026 |

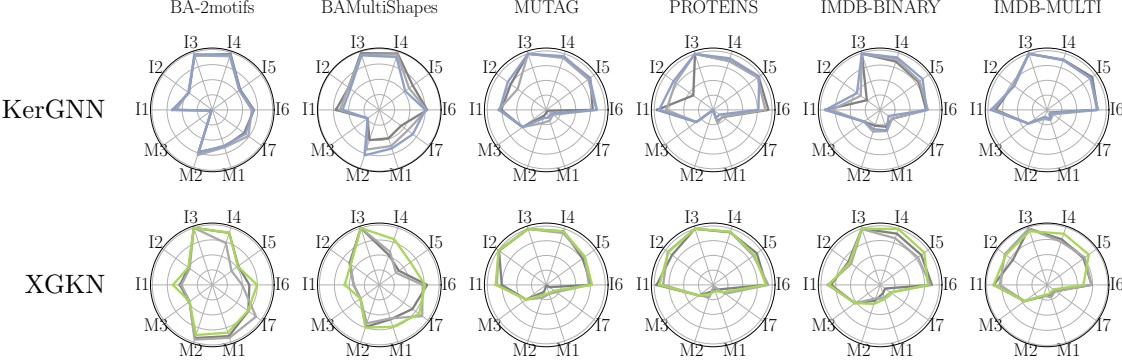

Figure 4: AIM metrics measured for KerGNN and XGKN with threshold: $p$, $p - 0.1$ in dark grey, and $p + 0.1$ in light grey (see threshold choice discussion in Section 6).

### A.5 Combined results

For easier readability and comparison, Table 7 and Table 8 show results achieved by XGKN and all baselines.

Table 7: Accuracy values; top-performing and statistically similar models are highlighted in bold.

|  | BA-2motifs | BAMultiShapes | MUTAG | PROTEINS | IMDB-B | IMDB-M |
|---|---|---|---|---|---|---|
| GAT | 92.4±6.4 | 90.0±9.7 | 80.0±9.9 | **73.7±3.5** | 69.8±3.7 | **45.1±3.2** |
| GIN | **99.6±0.7** | **94.5±3.0** | 81.5±9.8 | **75.4±4.1** | 70.0±4.2 | **45.0±5.1** |
| ProtGNN | **100.0±0.0** | 85.0±7.5 | **89.4±7.2** | 66.1±7.4 | **78.3±5.8** | **45.2±8.5** |
| GKNN | **99.2±0.2** | 90.7±7.9 | 75.7±6.9 | 68.5±6.6 | 61.1±9.4 | **44.1±4.2** |
| KerGNN | **99.4±0.1** | 90.4±4.9 | 81.9±6.0 | 70.7±8.0 | 71.1±7.4 | **43.7±4.5** |
| XGKN | **100.0±0.0** | 92.8±4.0 | 84.2±6.9 | **74.5±2.1** | 71.3±7.6 | **45.6±5.4** |

Table 8: Instance-level (A1) and model-level (A2) explanations accuracy scores; top-performing and statistically similar models are highlighted in bold

| | BA-2motifs | | BAMultiShapes | | MUTAG | |
|---|---|---|---|---|---|---|
| | A1 | A2 | A1 | A2 | A1 | A2 |
| GIN | **0.51±0.11** | – | **0.21±0.06** | – | 0.**78±0.03** | – |
| GAT | 0.26±0.10 | – | 0.17±0.05 | – | 0.73±0.06 | – |
| ProtGNN | 0.33±0.06 | 0.76±0.01 | **0.22±0.07** | 0.80±0.02 | 0.65±0.02 | 0.75±0.03 |
| GKNN | **0.49±0.04** | **0.80±0.01** | 0.17±0.11 | **0.83±0.01** | **0.80±0.03** | 0.77±0.01 |
| KerGNN | 0.33±0.12 | 0.68±0.01 | 0.10±0.06 | 0.72±0.01 | 0.76±0.04 | **0.80±0.01** |
| XGKN | **0.50±0.11** | **0.77±0.13** | **0.31±0.09** | 0.78±0.00 | **0.80±0.02** | **0.81±0.01** |

