# OpenReview forum: "AIMing for Standardised Explainability Evaluation in GNNs: A Framework and Case Study on Graph Kernel Networks"
_TMLR — Accepted by TMLR_

### Review · Reviewer_oCuC · 2026-02-07

**Summary Of Contributions:**

The paper introduces a new method called AIM for explainability in GNNs, which addresses the limitation of previous methods that primarily focus on post-hoc explanations. Specifically, the authors develop the SHAPExplainer, a method to extract instance-level explanations from inherently interpretable GNNs.

**Audience:**

Yes

**Audience Explanation:**

The paper addresses a critical gap in the field by providing a standardized evaluation framework that enables fair comparison across different explanation GNNs.

**Claims And Evidence:**

Yes

**Claims Explanation:**

The paper provides a thorough mathematical formulation of the AIM framework, along with detailed experimental validation across multiple datasets and multiple GNN models.

**Requested Changes:**

My suggestions are as follows:

1.I suggest the authors to strengthen the introductory section including the contributions of this work. This version omits many detail contributions of this paper, such as SHAPExplainer and  XGKN.

2.I have noticed that the authors report the running time cost in Table 3. So, Can you provide the complexity analysis of the proposed method?

3.One of the key contributions in this paper is the SHAPExplainer. Hence, a more detailed explanation of how it handles different types of GKNs would strengthen the contribution.

4.This paper focuses on inherently interpretable GNNs. Can you provide a more detailed discussion on how AIM could be applied to standard message-passing GNNs (like GIN) that don't have explicit interpretable components

---

> ### Author Response · Authors · 2026-02-17
>
> Thank you for the review, and in particular for your positive assessment of the relevance and thoroughness of our work.
>
> ### **1. Contributions**
> >I suggest the authors to strengthen the introductory section including the contributions of this work. This version omits many detail contributions of this paper, such as SHAPExplainer and XGKN.
>
> We have updated the introduction to include the contributions of SHAPExplainer and XGKN.
>
> ### **2. Complexity analysis**
> >I have noticed that the authors report the running time cost in Table 3. So, Can you provide the complexity analysis of the proposed method?
>
> The complexity of each method depends heavily on the hyperparameters of both the model and the explainer. The main computational cost of the SHAPExplainer in our framework arises from the forward pass of the Graph Kernel Network (GKN), which is the same as that of KerGNN and is discussed in more detail in [1].
>
> After computing the kernel similarities, aggregation and the final MLP have negligible cost.
>
> In contrast, iterative explainers such as IntegratedGradients or ShapleyValueSampling require multiple forward (and backward for IG) passes per explanation.
>
> Therefore, SHAPExplainer with GKN only requires a single forward pass per explanation, making it computationally more efficient than other explainers. This efficiency is reflected in the running times reported in Table 3.
>
> We note that generating explanations for ProtGNN with SHAPExplainer is slower. This is because ProtGNN operates on graph embeddings (not kernels) and sampled subgraphs. Consequently, each explanation requires evaluating a different set of subgraph samples, which increases the total runtime.
>
> [1] Feng, A., You, C., Wang, S., and Tassiulas, L. KerGNNs: Interpretable Graph Neural Networks with Graph Kernels, 2022.
>
>
> ### **3. SHAPExplainer for different types of GKNs**
> >One of the key contributions in this paper is the SHAPExplainer. Hence, a more detailed explanation of how it handles different types of GKNs would strengthen the contribution.
>
> The SHAPExplainer is derived assuming sum aggregation, while different GKNs may use other aggregation functions. For example:
>
> - **Max pooling:** If the aggregation selects the maximal element, $z_i = S_{ji}$ for the index $j$ that maximizes the value, the full attribution $\phi_i$ is assigned to that contributing element: $w_j = \phi_i$.
>
> - **Negative-entropy-style aggregation** (XGKN, Eq. 5): This can be written as $S_{ji} = \frac{R_{ji}}{\|R\|} \log \frac{R_{ji}}{\|R\|}$, which decomposes additively across components. Therefore, the standard sum-aggregation propagation can be applied.
>
> We include these cases and the corresponding propagation rules in the revised manuscript to clarify how SHAPExplainer generalizes to GKNs with different aggregation types (see Appendix A.1).
>
>
> ### **4. AIM for message-passing GNNs**
> >This paper focuses on inherently interpretable GNNs. Can you provide a more detailed discussion on how AIM could be applied to standard message-passing GNNs (like GIN) that don't have explicit interpretable components
>
> The AIM framework is general and can be applied to standard message-passing GNNs, such as GIN and GAT, as we demonstrate in our experiments. For these models, AIM evaluates interpretability for a given post-hoc explainer. Interestingly, we observe that GIN, despite lacking explicit interpretable components, achieves higher explainability scores than GAT, which uses attention mechanisms often assumed to provide interpretability.
>
> While AIM could in principle be used to improve such message-passing GNNs by identifying their shortcomings in the context of XAI, in this work we focus on using AIM as an evaluation framework and on Graph Kernel Networks (GKNs). GKNs allow AIM to probe model behavior at the level of concepts or motifs, providing a complementary perspective on interpretability that is less straightforward to study in standard message-passing GNNs.
>
>
> **We hope we resolved all your concerns, but let us know if there are any more questions.**

---

> > ### Comment · Reviewer_oCuC · 2026-04-01
> >
> > Thank you for your responses. I have no further questions.

---

### Review · Reviewer_mht6 · 2026-02-07

**Summary Of Contributions:**

The paper addresses a well-known gap in GNN XAI: most existing evaluation protocols are designed for post-hoc explainers applied to a fixed model, making it difficult to compare explanations across models and to systematically assess inherently interpretable GNNs. It proposes AIM, a standardized framework for evaluating explainability across Accuracy, Instance-level, and Model-level dimensions. The paper also introduces SHAPExplainer, tailored to Graph Kernel Networks (GKNs) (and claimed to extend to prototypical GNNs), and employs AIM-driven analysis to develop XGKN, a revised GKN architecture that improves explainability without sacrificing predictive performance.

**Additional Comments:**

I am providing some detailed explanations for the Requested Changes (C2-4):

**C2**: The authors claimed "Each metric ranges from 0 to 1" in Section 4.2.2, which contradicts the written formulas for M1-3:

* M1/M2: IoU is in $[0, 1]$ for one graph pair. If you sum IoU over many graphs in a dataset ($\mathcal{G} \in D$), you can easily get values bigger than 1 (e.g., 100 graphs × 0.5 IoU = 50). So as written, M1/M2 cannot be guaranteed to lie in $[0,1]$, contradicting the range claim.

* M3: Redundancy is defined via CORR between concept-response vectors. Standard correlation lies in $[-1, 1]$, So M3 could be negative, and it does not automatically live in $[0,1]$.

* Relatedly, the paper reports M1–M3 as $1-\gamma$ (to make “higher is better”), it is broken if $\gamma \notin [0, 1]$, and it could become negative (if $\gamma>1$) or exceed 1 (if $\gamma<0$).

**C3**: I am doubting M1/2's correctness. For I3/I4 (robustness), the paper explicitly compares explanations only under perturbations that keep the class prediction unchanged (with the condition $f(\mathcal{S})=c$). While for M1/M2, the paper perturbs the model’s concepts ($\mathcal{H}_i \to \mathcal{H}'_i$) and then compare $h(\mathcal{G})$ vs $h'(\mathcal{G})$ without any condition like “prediction stays the same.” If you change the model’s concepts enough, the model might predict a different class. In that case, it’s totally expected that the explanation changes, even if the instance/model-level alignment is poor. So M1/M2 might end up measuring “did the decision change?” rather than “did local explanations properly reflect concept edits?”

**C4**: The proposed SHAPExplainer computes SHAP values $\{\phi_i\}$ at the concept level (inputs $z$ to $f_{\text{pred}}$), and then “propagates” these attributions back to node/subgraph importance by reversing the aggregation step, explicitly assuming a summation aggregator $z_i=\sum_j S_{ji}$ (their derivation “for simplicity”). This assumption is not compatible with other aggregators used in the paper: ProtGNN is defined with max aggregation, and the proposed XGKN replaces aggregation with a nonlinear negative-entropy-style function (Eq. 5). In such cases, proportional redistribution can assign importance to nodes that have little/no effect on $z_i$, so the method becomes heuristic unless justified.

**Audience:**

Yes

**Audience Explanation:**

A unified explainability evaluation framework that supports cross-model comparisons, plus a case study on inherently interpretable GNNs (GKNs / prototype methods), is likely to be of interest to researchers working on GNN explainability, evaluation methodology, and interpretable ML.

**Broader Impact Concerns:**

No broader impact concerns.

**Claims And Evidence:**

No

**Claims Explanation:**

Mostly convincing/clear within the AIM + SHAPExplainer + XGKN story: the paper defines AIM metrics formally and ties them to explanation-quality properties, with a clear taxonomy of what is being measured at the instance and model level (Sec. 4, Table 2). The empirical utility/efficiency of SHAPExplainer and the claimed improvements of XGKN are also supported by experiments.

**C1**:  the paper searches $p \in \{0.1,\dots,0.9\}$ and chooses the one that maximizes A1 (with GT) or I1+I2 (without GT). If this is done on the same split used for reporting, it can inflate AIM scores since the threshold is effectively tuned using the evaluation objective. The paper does not clearly state a strict validation–test separation for this step. A clearer protocol (e.g., selecting thresholds on a validation set, reporting sensitivity to $p$, or using a fixed thresholding rule) would be needed to fully support the experimental claims.

**Requested Changes:**

### Critical changes:

**C1**: As described in C1 above.

**C2**: The paper claims each metric is in $[0,1]$ and that A2/M1–M3 are reported as $1-\gamma$, but the written formulas for M1/M2 appear unnormalised over $D$, and M3 uses CORR (typically $[-1,1]$). Please (i) fix the formulas (e.g., make M1/M2 averages/expectations), (ii) clarify what “CORR” means (Pearson? absolute correlation? shifted/scaled?), and (iii) correct all definitions to ensure the reported orientation $1-\gamma$ yields valid ranges.

**C3**: For I3/I4 you condition on perturbations that keep the predicted class unchanged, but for M1/M2 you perturb model concepts and compare explanations without a comparable condition. If concept edits change the predicted class, explanation changes are expected and may not indicate “local/global alignment.” Please either add a condition (e.g., evaluate only graphs where prediction is unchanged) or redefine the metric to be class-conditional / otherwise robust to label flips.

**C4**: The SHAPExplainer propagation step is derived assuming sum aggregation. This assumption is not compatible with (i) ProtGNN’s max aggregation, and (ii) XGKN’s nonlinear negative-entropy-style aggregation (Eq. 5). Either restrict claims/experiments to sum-like aggregators, or provide a correct derivation/justification (and potentially a modified propagation rule) for these aggregation types.

### Strengthening changes:

**C5**: Improve Fig. 1 make it clearer and easier to follow.

---

> ### Author Response · Authors · 2026-02-17
>
> We thank the reviewer for their thoughtful and thorough feedback, and for recognizing the relevance of our work. The comments and suggestions have helped us improve the clarity and quality of the manuscript.
>
> ### C1. Validation–test separation
> >C1: the paper searches $p\in 0.1,...,0.9$ and chooses the one that maximizes A1 (with GT) or I1+I2 (without GT). If this is done on the same split used for reporting, it can inflate AIM scores since the threshold is effectively tuned using the evaluation objective. The paper does not clearly state a strict validation–test separation for this step. A clearer protocol (e.g., selecting thresholds on a validation set, reporting sensitivity to $p$, or using a fixed thresholding rule) would be needed to fully support the experimental claims.
>
> Thanks for raising this.
> We do indeed search for the best threshold on what we report on.
> Our reasoning for this was based on the fact that in most cases, one is not likely to have access to labels (as with A1), which would necessitate use of some intrinsinc measure that is likely computable on new data.
>
> However we do take the point that this setup could be introducing some leakage.
> To evaluate this, we perform an experiment where the threshold $p$ is chosen based on evaluation on a validation split, and the metric is reported on both the validation and held-out test sets. Results, shown in tables below, indicate that the difference between tuning on validation versus test is quite small.
>
> The tables below report the computed metrics in the format *validation score / test score* (mean + standard deviation).
>
> |Model  |Metric |BA-2motifs         |BAMultiShapes      |MUTAG              |
> |-------|-------|-------------------|-------------------|-------------------|
> |GIN    |A1     |0.51±0.15/0.51±0.16|0.21±0.05/0.22±0.06|0.79±0.07/0.78±0.06|
> |GAT    |A1     |0.28±0.30/0.27±0.19|0.20±0.32/0.17±0.21|0.77±0.06/0.74±0.07|
> |ProtGNN|A1     |0.37±0.13/0.32±0.07|0.22±0.05/0.24±0.09|0.79±0.01/0.77±0.02|
> |GKNN   |A1     |0.50±0.01/0.50±0.02|0.17±0.03/0.17±0.05|0.81±0.01/0.80±0.02|
> |KerGNN |A1     |0.32±0.06/0.31±0.04|0.12±0.02/0.11±0.01|0.74±0.04/0.74±0.04|
> |XGKN   |A1     |0.51±0.07/0.50±0.08|0.29±0.06/0.29±0.07|0.80±0.03/0.79±0.03|
>
> |Model  |Metric|PROTEINS           |IMDB-BINARY        |IMDB-MULTI         |
> |-------|------|-------------------|-------------------|-------------------|
> |GIN    |I1    |0.94±0.03/0.92±0.05|0.92±0.07/0.91±0.07|0.90±0.05/0.89±0.05|
> |GIN    |I2    |0.77±0.07/0.78±0.08|0.74±0.10/0.74±0.07|0.77±0.15/0.71±0.16|
> |GAT    |I1    |0.94±0.03/0.94±0.04|0.96±0.03/0.96±0.03|0.95±0.04/0.94±0.04|
> |GAT    |I2    |0.78±0.12/0.78±0.10|0.51±0.17/0.47±0.16|0.64±0.17/0.64±0.15|
> |ProtGNN|I1    |0.97±0.05/0.92±0.11|0.86±0.02/0.93±0.02|0.92±0.08/0.91±0.06|
> |ProtGNN|I2    |0.85±0.13/0.85±0.05|0.60±0.07/0.58±0.03|0.92±0.05/0.79±0.07|
> |GKNN   |I1    |0.92±0.03/0.92±0.03|0.97±0.04/0.98±0.01|0.96±0.01/0.95±0.00|
> |GKNN   |I2    |0.64±0.11/0.67±0.17|0.70±0.24/0.71±0.14|0.67±0.18/0.66±0.14|
> |KerGNN |I1    |0.84±0.05/0.83±0.02|0.93±0.06/0.91±0.02|0.98±0.01/0.96±0.03|
> |KerGNN |I2    |0.82±0.04/0.82±0.05|0.47±0.18/0.46±0.08|0.74±0.18/0.74±0.16|
> |XGKN   |I1    |0.94±0.03/0.94±0.02|0.95±0.03/0.94±0.03|0.95±0.04/0.94±0.04|
> |XGKN   |I2    |0.99±0.02/0.97±0.01|0.70±0.22/0.71±0.11|0.85±0.11/0.83±0.14|
>
>
> In our approach, the threshold is selected based on at most 1–2 metrics and then applied uniformly across all metrics, which further reduces the risk of overfitting.
> We believe the updated results show that the methods are not sensitive to the hyperparameter search for $p$.
>
>
> ### C2. Metrics formulations
> > The paper claims each metric is in $[0,1]$ and that A2/M1–M3 are reported as $1-\gamma$, but the written formulas for M1/M2 appear unnormalised over
> , and M3 uses CORR (typically
> ). Please (i) fix the formulas (e.g., make M1/M2 averages/expectations), (ii) clarify what “CORR” means (Pearson? absolute correlation? shifted/scaled?), and (iii) correct all definitions to ensure the reported orientation
>  yields valid ranges.
>
> You are correct; M1/M2 should be shown as normalized by $|D|$, i.e., expressed as averages over the dataset.
>
> For M3, $\mathrm{CORR}$ denotes the absolute Spearman rank correlation. We use Spearman correlation because the compared responses are subsequently processed by an MLP, making rank-based (monotonic) agreement more appropriate than linear correlation. The absolute value ensures the metric lies in $[0,1]$ and captures alignment strength irrespective of direction.
>
> We revised the definitions in the updated manuscript to ensure these metrics are properly normalized and consistent with the reported $1-\gamma$ orientation and $[0,1]$ range.
>
> **Responses to the remaining comments are provided in the second comment below.**

---

> > ### Author Response · Authors · 2026-02-17
> >
> > ### C3. Perturbations in M1/M2
> > >For I3/I4 you condition on perturbations that keep the predicted class unchanged, but for M1/M2 you perturb model concepts and compare explanations without a comparable condition. If concept edits change the predicted class, explanation changes are expected and may not indicate “local/global alignment.” Please either add a condition (e.g., evaluate only graphs where prediction is unchanged) or redefine the metric to be class-conditional / otherwise robust to label flips.
> > >If you change the model’s concepts enough, the model might predict a different class. In that case, it’s totally expected that the explanation changes, even if the instance/model-level alignment is poor. So M1/M2 might end up measuring “did the decision change?” rather than “did local explanations properly reflect concept edits?”
> >
> > In principle, changes in the predicted class can induce changes in the explanation, and that conditioning on label stability is important for certain types of perturbation analyses (as done for I3/I4).
> >
> > However, M1/M2 are designed to evaluate a different notion of alignment. In contrast to I3/I4, which assess explanation robustness under input perturbations, M1/M2 quantify whether explanations reflect model-level concept interventions. Specifically, we modify a given motif (concept) at the model level (model perturbation), and measure whether this intervention affects instance-level explanations.
> >
> > Importantly, the perturbed model is applied to the entire dataset. Conditioning on instances where the predicted class remains unchanged would substantially restrict the evaluation set and introduce selection effects, since concept edits may legitimately alter decision boundaries. In fact, a change in prediction can be a meaningful consequence of altering concepts. From this perspective, explanation changes following concept edits are not inherently undesirable; rather, they are expected if explanations faithfully reflect the modified model behavior.
> >
> > To clarify this point and address the reviewer’s concern, we report the percentage of instances for which the predicted label does not change under model perturbations in M1/M2 in the table below.
> >
> > |Model  |Metric|BA-2motifs|BAMultiShapes|MUTAG    |PROTEINS |IMDB-BINARY|IMDB-MULTI|
> > |-------|------|----------|-------------|---------|---------|-----------|----------|
> > |ProtGNN|M1    |1.00±0.00 |1.00±0.00    |0.60±0.20|0.70±0.34|0.94±0.07  |0.62±0.37 |
> > |ProtGNN|M2    |0.33±0.47 |0.61±0.44    |0.38±0.30|0.44±0.35|0.60±0.15  |0.64±0.11 |
> > |GKNN   |M1    |0.67±0.23 |0.59±0.17    |0.75±0.16|0.68±0.15|1.00±0.00  |1.00±0.00 |
> > |GKNN   |M2    |0.50±0.02 |0.53±0.06    |0.66±0.05|0.72±0.13|0.56±0.23  |0.56±0.12 |
> > |KerGNN |M1    |0.50±0.02 |0.40±0.05    |0.80±0.16|0.81±0.27|0.56±0.08  |0.18±0.06 |
> > |KerGNN |M2    |0.50±0.02 |0.40±0.05    |0.74±0.17|0.93±0.12|0.57±0.11  |0.33±0.08 |
> > |XGKN   |M1    |0.51±0.09 |0.53±0.12    |0.55±0.06|0.80±0.21|0.65±0.17  |0.39±0.21 |
> > |XGKN   |M2    |0.59±0.16 |0.40±0.07    |0.45±0.06|0.30±0.09|0.39±0.12  |0.33±0.25 |
> >
> >
> > ### C4. SHAPExplainer for different aggregation types
> > >The SHAPExplainer propagation step is derived assuming sum aggregation. This assumption is not compatible with (i) ProtGNN’s max aggregation, and (ii) XGKN’s nonlinear negative-entropy-style aggregation (Eq. 5). Either restrict claims/experiments to sum-like aggregators, or provide a correct derivation/justification (and potentially a modified propagation rule) for these aggregation types.
> >
> > When max pooling is used (e.g., in ProtGNN), the aggregated representation satisfies $z_i=S_{ji}$ for the maximising index $j$. In this case, we assign the full attribution $\phi_i$ to the contributing element $j$, i.e. $w_j=\phi_i$.
> >
> > For negative entropy (e.g., in XGKN), the aggregation can be written as $S_{ji}=\frac{R_{ji}}{||R||}log \frac{R_{ji}}{||R||}$, which decomposes additively across components. This allows us to apply the same propagation rule as in the sum-aggregation case.
> >
> > We have included these derivations in the revised manuscript.
> >
> > ### C5. Diagram
> > >Improve Fig. 1 make it clearer and easier to follow.
> >
> > We have updated the figure to improve clarity regarding its components and their separation.
> >
> > **We hope we resolved all your concerns, but let us know if there are any more questions.**

---

> > > ### Comment · Reviewer_mht6 · 2026-03-19
> > >
> > > Thanks for the revision and clarity; my concern has been addressed.

---

### Review · Reviewer_kmxP · 2026-02-25

**Summary Of Contributions:**

This paper proposes AIM, a unified evaluation framework for explainability in Graph Neural Networks (GNNs). Unlike prior frameworks that mainly evaluate post-hoc explainers on a fixed model, AIM enables cross-model comparison and is applicable to inherently interpretable architectures such as Graph Kernel Networks (GKNs) and Prototypical GNNs. The framework evaluates explainability along three dimensions: (1) Accuracy: alignment with ground-truth explanations. (2) Instance-level explanations: sufficiency, necessity, robustness, and consistency. (3) Model-level explanations: correctness and redundancy. Based on the evaluation dimensions, the paper further introduces a SHAPExplainer tailored for GKNs, propagating SHAP values through kernel responses. A case study showing that AIM-based insights lead to an improved model, XGKN, which enhances explainability without sacrificing predictive accuracy.

**Audience:**

Yes

**Audience Explanation:**

The paper addresses a central issue in explainable AI for graph learning. Researchers working on GNN interpretability, evaluation protocols for XAI, inherently interpretable neural architectures, and graph kernel methods would find the framework valuable.

**Broader Impact Concerns:**

The work is methodological and evaluation-focused. It promotes better transparency in GNN systems and does not introduce direct ethical risks. No major broader impact concerns are identified.

**Claims And Evidence:**

Yes

**Claims Explanation:**

The experimental design is thorough, covering multiple datasets and multiple model types. The paper includes ground-truth evaluation where available, statistical significance testing, comparison against strong baselines, and runtime analysis for explainers. However, additional sensitivity analyses on thresholding and metric interactions would further strengthen the claims.

Strengths:
S1. The paper convincingly identifies a real gap in GNN explainability evaluation, especially the lack of cross-model comparability. Unlike many prior works, AIM is not restricted to post-hoc explainers applied to a single model.
S2. The AIM framework integrates multiple well-established explainability properties into a unified evaluation protocol.
S3. The analysis reveals that interpretable components do not necessarily produce faithful explanations. The redundancy and weak correctness of model-level explanations are important observations.

Weaknesses:
W1. The choice of certain metrics, especially model-level correctness (M1–M2), lacks deeper theoretical grounding.
W2. The percentile-based thresholding strategy may influence results significantly. A sensitivity analysis would strengthen the claims.
W3. The architectural improvement (XGKN) is limited to GKN-style models. Broader generalization remains speculative.

**Requested Changes:**

Major request
1.	Add sensitivity analysis for thresholding parameter selection.
2.	Clarify theoretical intuition behind model-level correctness metrics.

Suggested Improvements
1.	Expand discussion on generalization beyond GKN-based models.
2.	Include more discussion on potential real-world deployment implications.
3.	Improve clarity of some formulas and explanations in Sections 4 and 5.

---

> ### Author Response · Authors · 2026-02-28
>
> We thank the reviewer for their encouraging feedback and for acknowledging the contributions of AIM to GNN explainability, as well as our thorough experiments and insights. We address the suggested points and clarifications below.
>
> ## MR1. Sensitivity Analysis
> >Add sensitivity analysis for thresholding parameter selection.
>
> We have provided a simple sensitivity test in Appendix A.4, to demonstrate that our selection of the threshold, while clearly relevant to obtaining the reported results, does not mean that there is some limited special value that makes things work.
>
> Evaluations of KerGNN and XGKN under perturbations of $\pm 0.1$ from the optimal threshold $p$ demonstrate consistent performance stability. The observation that these shifts overall result in similar or slightly worse performance confirms the robustness of the original selection.
>
> ## MR2. Model-level Correctness
> >Clarify theoretical intuition behind model-level correctness metrics.
>
> The intuition for these metrics (M1, M2) is simply that the interpretable component of the model (graph filters, prototypes, etc.), which distils consistent patterns of information across the dataset, when perturbed _should_ change/affect the instance-level explanations obtained with this changed model.
>
> If the model learns a filter that captures 'rings' in graphs, editing this to _break_ the ring filter, say, should materially affect the decision making and subsequent explanation derived for any model that has learnt to use this 'ring' filter correctly.
>
> Note that while I3 and I4 evaluate explanation robustness under input perturbations (altering the input graph itself), M1 and M2 evaluate alignment under model perturbations (altering the model's internal concepts). M1 and M2 quantify this transparency. If instance-level explanations remain unchanged after model-level concept edits, it suggests the model may not be utilising those learned concepts as intended.
>
> ## SI1. Generalisation
> >Expand discussion on generalization beyond GKN-based models.
> >The architectural improvement (XGKN) is limited to GKN-style models. Broader generalization remains speculative.
>
> While the architectural improvements in XGKN are specific to Graph Kernel Networks, the AIM framework itself is formulated with minimal constraints to facilitate broad applicability across any GNN architecture. We demonstrate this through experiments on standard message-passing GNNs such as GIN and GAT.
>
> The primary value of AIM lies in its diagnostic ability to identify specific weaknesses, such as low robustness or high redundancy, which allows researchers to drive improvements in any architecture. The underlying principles we applied to GKNs can generalise naturally to other similar graph models, such as Prototypical GNNs.
>
> Notably, our evaluation showed that GAT, despite its attention-based interpretability mechanism, produced weaker explanations than GIN, showing that theoretical interpretability does not always carry over in practice.
>
>
> ## SI2. Real-world deployment
> >Include more discussion on potential real-world deployment implications.
>
> We have added a discussion addressing the potential implications of AIM for real-world deployment to Section 4.1.
>
> The AIM framework provides a standardised suite of *sanity checks*&mdash;
> measuring properties such as Consistency and Necessity&mdash;
> that allow developers to quantify the reliability of an XAI system before deployment. AIM can help identify specific system weaknesses, particularly when different desirable explainability properties may conflict. For example, our experiments reveal that inherent interpretability can be misleading if the model does not utilise its components faithfully, a factor that must be evaluated rigorously before deployment in high-stakes environments such as molecular analysis or fraud detection.
>
>
> ## SI3. Formulas and Explanations Clarity
> >Improve clarity of some formulas and explanations in Sections 4 and 5.
>
> We have revised Sections 4 and 5 to improve technical clarity based on reviewers' feedback. These edits were made to address concerns regarding mathematical consistency and clarity. If there is any particular aspect of the formulation you would like further refinement on, please do let us know.
>
>
> **We hope we resolved all your concerns, but let us know if there are any more questions.**

---

### Decision · Action_Editor_fdCJ · 2026-04-04

**Recommendation:** Accept as is

**Audience:**

Yes

**Audience Explanation:**

The work targets an active area at the intersection of explainable AI and graph learning. Researchers working on GNN interpretability, evaluation methodology, and inherently interpretable architectures will find the proposed framework and empirical insights useful.

**Claims And Evidence:**

Yes

**Claims Explanation:**

The paper introduces AIM, a unified and well-motivated framework for evaluating explainability in GNNs, addressing a clear gap in current practice where comparisons across models and explanation types are difficult. The experimental study is thorough, covering multiple datasets and model classes. Also, both instance-level and model-level metrics are provided. In the discussion phase, the authors largely addressed concerns from the reviewers, including metric definitions, normalization issues, and additional analyses (e.g., sensitivity and validation-test separation). The case study on GKNs and the improved XGKN model further demonstrate the practical use of the framework.